# IMPROVING GENERALIZATION BY LOSS MODIFICATION

**Michael Tetelman**
BayzAI.com, Volkswagen Group of America, IECC
`michael.tetelman@gmail.com`

## ABSTRACT

What data points from available data set should be used for training? For all subsets of available data it will generally make different solutions. We show that a simple loss modification allows to find a single solution that represents data set properties and not particular selections of data points thus improving the generalization performance.

The purpose of training neural networks is to find a solution that is able to make accurate predictions for new never seen before data samples that supposedly obtained from the same distribution as training data. To achieve that neural networks must learn properties of the distribution of available data and not specific details of data samples in the data set.

For a given data set of $N$ data samples the usual approach is to use all it for training. However, we can ask ourselves, what is special about that $N$? What if we select a subset of available data for training a neural network. We expect that all random subsets will share the same statistical properties. Nevertheless, as every practitioner knows, training a neural network with any selected subset of a larger data set will produce a different solution.

Can we find a single solution that depends on distributional properties of a data set and not on a particular selection of data samples for training? The answer is yes, it is possible.

We will consider Bayesian approach for learning a neural network solution (Wilson & Izmailov, 2020) that is defined as following. Let's $D$ be a data set of $N$ data samples and $\{D_i\}$ are all possible subsets of $D$ indexed by $i$. For every data sample $n$ with input $x_n$ and label $y_n$ the model computes a probability $P(y_n|x_n, w)$ parameterized by weights $w$.

Then probability distribution of weights for the model trained on data set $D_i$ is

$$P(w|D_i) = \prod_{n \in D_i} P(y_n|x_n, w)P_0(w)/P(D_i). \tag{1}$$

Here, $P_0(w)$ is a prior probability distribution of weights and normalization factor $P(D_i)$ is probability of data set $D_i$. The top factor in the ratio is a probability of data set $D_i$ for given weights $P(D_i|w)$ times prior of weights $P_0(w)$, so

$$P(D_i|w) = \prod_{n \in D_i} P(y_n|x_n, w). \tag{2}$$

The solution $w_i$ for data set $D_i$ is found by maximizing the probability distribution of weights: $w_i = \arg\max_w P(w|D_i)$, where $P(w|D_i) \propto P(D_i|w)P_0(w)$. Obviously it depends on the data in $D_i$ and it's different for every data set.

We will use the following principle: if we do not know how to select a best data set from available subsets we should average over all of them.

Then to find a solution that is not specific to a data set we can use a weight distribution that is averaged by probabilities of all data sets

$$P(w|\{D_i\}) = \frac{\sum_i P(w|D_i)P(D_i)}{\sum_i P(D_i)}. \tag{3}$$

With the definition in eq.2 the average weight distribution can be rewritten as follows

$$P(w|\{D_i\}) = P_0(w) \sum_i P(D_i|w) / \sum_i P(D_i). \tag{4}$$

Now, the sum of probabilities of all subsets of the N-point data set, including an empty one, for given weights is equal to the following expression

$$\sum_i P(D_i|w) = \prod_{n \in D} \left(1 + P(y_n|x_n, w)\right). \tag{5}$$

By using losses per data sample $l_n(w) = -\log P(y_n|x_n, w)$ the total loss based on the average weight distribution is given by

$$L = -\sum_{n=1}^{N} \log \left(1 + e^{-l_n(w)}\right) - \log P_0(w) \tag{6}$$

Compared with the usual loss definition $L_0(w) = \sum_{n=1}^{N} l_n(w) - \log P_0(w)$ the loss in the eq.6 has contributions of high-loss data samples suppressed. Suppressing outliers for improving convergence is a well known heuristics. The general example of that is to use absolute linear loss or the Huber loss (Huber, 1964). Another approach based on heuristics is to select low-loss samples to improve robustness of SGD (Shah et al., 2020).

Unlike these heuristics our approach is based on a principle of using a weight distribution by averaging it over all possible choices of data selection. The result is the outlier suppression loss in the eq.6. While the outlier suppression loss gradient per sample is proportional to the original gradient it has a suppression factor essentially removing the outlier contributions to a total gradient

$$\partial_w \left[-\log\left(1 + e^{-l_n(w)}\right)\right] = \frac{\partial_w l_n}{1 + e^{l_n}} \tag{7}$$

The format of this paper does not allow to provide much of experimental results. However, we can mention that experiments show that solutions with the generalized loss in the eq.6 have at least 10% improvements in the prediction accuracy for all tested problems. Also, the outlier suppression loss was used for training GANs with BCE loss effectively stabilizing convergence to Nash equilibrium.

URM STATEMENT

The author acknowledge that the author of this work meets the URM criteria of ICLR 2023 Tiny Papers Track.

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
