# OpenReview forum: "Improving generalization by loss modification"
_ICLR.cc/2023/TinyPapers — Submitted to Tiny Papers @ ICLR 2023_

### Official Review · Reviewer_6pCj · 2023-03-30

**Confidence:** 5

**Summary Of Contributions:**

The paper presents an important topic in generalization of model performance by 'differential' parameter optimization. Parameters of a neural network are learned by averaging its distribution over all possible subsets of the training set.

**Rating:**

Great Start (GS): a submission which meets some of the reviewing criteria but has room for improvement

**Strengths And Weaknesses:**

1. The topic dealt with in the paper is of utmost significance :- improving the generalization power of neural networks.
2. The new loss function that has been proposed exhibits insensitivity to high-loss data points. Thus, the model is provably robust to outliers and hence generalizes well.
3. The structure of the paper is mostly monolithic. It lacks in a specific section for concluding remarks and future work as well as some insights on the experiments performed and comparison study with state-of-the-art learning techniques.

**Suggested Changes:**

I thought that the paper highlights an important area in learning theory of neural nets, however providing a structure to the paper by adding sections would be nice. It would also help in reproducing the results of the paper if some insights on experimental results is provided.

---

### Official Review · Reviewer_qxp4 · 2023-03-30

**Confidence:** 3

**Summary Of Contributions:**

This paper proposes a simple loss modification to improve generalization in neural networks by considering the properties of the data set distribution rather than specific data samples.

**Rating:**

Needs Clarification (NC): a submission which does not meet the reviewing criteria and needs clarification for its described problem or solution

**Strengths And Weaknesses:**

Strengths
1. This paper is well written, motivation is clear and I enjoyed reading it.
2. The loss modification method is simple and can be easily implemented.

Weakness
1. No experimental Results
2. Not connected well to existing literature.

**Suggested Changes:**

Suggested Changes:
1. Incorporate atleast 1 set of experiments to show the validity of your method.
2 Connect with the existing literature.

---

### Official Review · Reviewer_LUHz · 2023-04-01

**Confidence:** 4

**Summary Of Contributions:**

The paper proposes a loss modification approach to improve generalization performance in neural networks.

**Rating:**

Great Start (GS): a submission which meets some of the reviewing criteria but has room for improvement

**Strengths And Weaknesses:**

The authors provide a sound theoretical framework for their approach. However, the paper could benefit from an empirical evaluation of the proposed approach to demonstrate its effectiveness. Additionally, the paper does not include a description of the data used in the experiments, making it difficult to replicate the results.

**Suggested Changes:**

- The authors should consider rephrasing the abstract and introduction to better highlight the main contribution of the paper.
- The paper would benefit from a more detailed description of the experiments, including the data sets used, the choice of hyperparameters, and the evaluation metrics used to demonstrate the effectiveness of the proposed approach.

---

### Comment · Program_Chairs · 2023-06-15
**Please give consent on archival**

Dear authors,

Your revision is accepted, however we do not see a consent given that this submission can be archived. Please do so ASAP.

-PCs

---

> ### Author Response · Authors · 2023-06-16
> **Consent on archival**
>
> I would like to opt-in for archival of the paper.

---

### Meta-Review · Area_Chair_YWQn · 2023-04-05

**Recommendation:** Invite to revise
**Confidence:** 5

**Metareview:**

The paper addresses the crucial topic of enhancing neural network generalization through a new loss function. However, in its current form it lacks experimental results to justify the modification in loss function.

**Summary:**

The paper proposes a loss modification approach to improve generalization performance in neural networks.

**Reason For Not Giving A Higher Recommendation:**

The paper currently lacks descriptions of the data and methodologies used in the experiments, which hinders its reproducibility.

**Reason For Not Giving A Lower Recommendation:**

NA

---

### Decision · Program_Chairs · 2023-04-08

Revision accepted; invite to archive

---

> ### Author Response · Authors · 2023-06-16
> **Thank you**
>
> Dear Program Chairs and Reviewers,
>
> Thank you for the reviews and for your decision.
>
> Sincerely,
> MT